# Friend or Foe: Paradoxical Roles of Autophagy in Gliomagenesis

**DOI:** 10.3390/cells10061411

**Published:** 2021-06-06

**Authors:** Don Carlo Ramos Batara, Moon-Chang Choi, Hyeon-Uk Shin, Hyunggee Kim, Sung-Hak Kim

**Affiliations:** 1Department of Animal Science, College of Agriculture and Life Sciences, Chonnam National University, Gwangju 61186, Korea; don_laze@yahoo.com (D.C.R.B.); sinhu97@gmail.com (H.-U.S.); 2Department of Biomedical Science, Chosun University, Gwangju 61452, Korea; choist777@chosun.ac.kr; 3Department of Biotechnology, College of Life Sciences and Biotechnology, Korea University, Seoul 02841, Korea; hg-kim@korea.ac.kr

**Keywords:** glioblastoma multiforme, autophagy, treatment

## Abstract

Glioblastoma multiforme (GBM) is the most common and aggressive type of primary brain tumor in adults, with a poor median survival of approximately 15 months after diagnosis. Despite several decades of intensive research on its cancer biology, treatment for GBM remains a challenge. Autophagy, a fundamental homeostatic mechanism, is responsible for degrading and recycling damaged or defective cellular components. It plays a paradoxical role in GBM by either promoting or suppressing tumor growth depending on the cellular context. A thorough understanding of autophagy’s pleiotropic roles is needed to develop potential therapeutic strategies for GBM. In this paper, we discussed molecular mechanisms and biphasic functions of autophagy in gliomagenesis. We also provided a summary of treatments for GBM, emphasizing the importance of autophagy as a promising molecular target for treating GBM.

## 1. Introduction

Glioblastoma multiforme (GBM) is the most malignant and common type of infiltrative glioma, a group of primary tumors arising from the central nervous system (CNS) [1,2]. GBM is categorized by the World Health Organization (WHO) as a grade IV astrocytoma because of its aggressive and highly proliferative nature [1]. Although GBM is a rare type of tumor with a global incidence of less than 10 per 100,000 people, it has become a major public health concern because of its poor prognosis [3]. Surgical resection accompanied by chemoradiation is currently the standard therapy for GBM [4]. Temozolomide (TMZ) is an oral alkylating drug most commonly used to treat GBM and astrocytomas. It can trigger cell cycle arrest at the G2/M phase, leading to apoptotic cell death [5]. It has been shown that the median overall survival of GBM patients treated with a combination of TMZ and irradiation (IR) can be improved to 14.6 months as compared to 12.1 months with IR alone [6]. However, not all GBM patients can respond to these treatments. Some GBM patients frequently display innate or acquired chemoresistance that can lead to tumor recurrence [5].

Isocitrate dehydrogenase 1/2 (IDH1/IDH2) mutation and methylation of the O^6^-methylguanine-DNA methyltransferase (MGMT) promoter have been found in GBM [7]. IDH mutations are commonly observed in patients with lower-grade gliomas and secondary GBM. Patients with IDH mutation have neomorphic enzyme activity that can transform α-ketoglutarate to D-2-hydroxyglutarate, which has implications in cell metabolism [8]. MGMT is a DNA repair enzyme that also plays a key role in GBM’s resistance to treatment [6]. Methylation of the promoter region of MGMT can lead to epigenetic silencing of the MGMT gene and confer TMZ sensitivity [7]. As a result, patients with methylated MGMT exhibit compromised DNA repair mechanisms. When the MGMT enzyme is turned on, it interferes with the effects of treatment [9]. Important genetic events in GBM include the following: (1) amplification and mutational activation of receptor tyrosine kinase (RTK) genes leading to abnormal growth factor signaling; (2) hyperactivation of phosphatidylinositol-3-OH kinase (PI3K) pathway; and (3) deficiency of p53 and retinoblastoma tumor suppressor pathways [10].

Autophagy has become a major target for drug development [11], as it plays an important role in GBM pathogenesis [12]. Autophagy is a highly conserved homeostatic mechanism that degrades and recycles intracellular components, such as damaged organelles, misfolded proteins, macromolecules, and foreign materials. It is coined from the Greek words “auto” and “phagy,” meaning “self-eating.” During the process, it requires the creation of an autophagosome, a double-membrane structure containing cytoplasmic materials that are sequestered and eventually fused with the lysosome for recycling and degradation [13,14]. Autophagy is normally activated for nutrient supply during cell development. It is involved in intracellular protein regulation and organelle turnover to maintain homeostasis [15]. Autophagy plays a paradoxical role as it can promote or suppress GBM. Several studies have shown that autophagy can either promote or inhibit tumor growth depending on unknown characteristics of numerous types of cancer [16]. Autophagy can promote cancer development by recycling intracellular substrates to maintain metabolic activity and mitochondrial function [17]. When tumor cells are deprived of growth factors, nutrients, and oxygen, they can sustain their survival through autophagy by degrading misfolded and damaged proteins or cellular components [18]. This allows them to survive and resume proliferation and initiation, contributing to their acquisition of resistance to treatments [19,20]. On the other hand, autophagy can suppress GBM by removing toxic unfolded proteins, oncogenic protein substrates, chromosomal instability, and damaged organelles [17].

However, molecular mechanisms involved in the “double-edged sword” role of autophagy in GBM remain unclear. It is important to understand the function of autophagy in GBM as well as its molecular mechanisms and underlying signaling pathways that regulate cellar death and survival. Thus, the objective of this paper was to discuss molecular mechanisms involved in the ability of autophagy to promote or suppress GBM. We also examined therapeutic methods and approaches for modulating autophagy activation or inhibition in GBM therapy.

## 2. Molecular Mechanisms of Autophagy

Autophagy is a physiological mechanism that acts as an adaptive cell response to stimuli or stresses to maintain cellular metabolism and homeostasis. When cells are deprived of nutrients, autophagy can degrade and recycle macromolecules to synthesize essential components and to provide energy supply [21,22]. Moreover, autophagy can serve as a recycling system to degrade and eliminate intracellular aggregates and damaged organelles to maintain metabolic homeostasis. This prevents the accumulation of toxic components and eliminates intracellular pathogens, allowing cells to adapt to environmental changes [23]. Autophagy is also controlled during hypoxia to reduce oxidative stress caused by low oxygen levels [21].

### 2.1. Types of Autophagy

Autophagy is carried through three main pathways (Figure 1): (Figure 1A) *Macroautophagy*, a process of encasing cellular contents and proteins in special structures called autophagosomes. After fusing with the lysosome, the autophagosome delivers its contents into the lumen for degradation; (Figure 1B) *Microautophagy*, a process that destroys cytoplasmic molecules by directly invaginating the lysosomal membrane; and (Figure 1C) *Chaperone-mediated autophagy (CMA)*, a selective mechanism in which heat shock protein 70 (HSC70) acts as a molecular chaperone that can recognize unique substrates containing the KFERQ motif and transport the protein cargo to the membrane of the lysosome, where it is bound to the cytolytic tail of lysosomal-associated membrane protein type 2 A (LAMP2A), resulting in degradation [11,24,25,26].

Selective autophagy has recently been considered as a highly regulated and specialized autophagic pathway for eliminating unwanted cytolytic components, such as damaged organelles, protein aggregates, and intracellular pathogens, which necessitate molecular machinery to ensure cargo identification and sequestration within autophagosomes [27,28,29]. Ubiquitin (Ub) and ubiquitin-like molecular modifiers can label damaged, defective, and unwanted organelles for degradation. These post-translational modifications (PTMs) can connect ubiquitinated cargoes with microtubule-associated protein 1B-light chain 3B (LC3B) so that they can be recognized by autophagy cargo receptors. The receptor cargo complex is engulfed by the autophagosome and delivered to the lysosome for degradation [30]. The different forms of selective autophagy or mammalian Ub-mediated organellophagy are outlined in Table 1.

Macroautophagy is the cell’s main and most common mechanism for removing damaged organelles. Therefore, this paper will focus on macroautophagy, which will be referred to as autophagy henceforth.

### 2.2. The Autophagic Pathway

Autophagy encompasses the following stages: (1) induction; (2) nucleation; (3) elongation and maturation; (4) fusion with lysosome; and (5) degradation and recycling (Figure 2) [26,35]. Autophagy induction includes the formation and expansion of a double-membrane vesicle called a phagophore [36]. This is followed by nucleation and elongation where the target autophagic substrate is sequestered and the phagophore becomes elongated. During the maturation stage, the phagophore extends around the entrapped substrate, forming an autophagosome. In the fusion stage, the mature autophagosome joins the lysosome to form an autophagolysosome. Finally, lysosomal enzymes can degrade the substrate and the inner membrane of an autophagosome, resulting in autolysosome [37]. Catabolic products formed are used in a variety of metabolic processes before they are degraded to provide ATP [35].

Autophagy at the basal level is regulated by a complex series of signaling events that are induced by various stimuli and cellular stresses [38]. The presence of growth factors and nutrients is detected by the mechanistic target of rapamycin (mTOR), a highly conserved Ser/Thr kinase that exists in two complexes (mTORC1 and mTORC2), each having its function and localization [21,39]. AMP-activated protein kinase (AMPK) controls the activity of mTORC1. Under homeostatic conditions, PI3K/protein kinase B (AKT) signaling activates mTORC1, inhibiting the autophagic mechanism by phosphorylating Unc-51like autophagy activating kinase 1/2 (ULK1/2) [40,41]. During nutrient deprivation or genotoxic stress, AMPK is increased, which inhibits the activity of mTORC1 and induces the autophagic process [42,43]. The ULK1 complex includes ULK1/2, FIP200, ATG10, and ATG13. It is then phosphorylated by AMPK, allowing it to migrate from the cytosol to the ER membrane [40,41,42]. RTKs, including epidermal growth factor receptor (EGFR), vascular endothelial growth factor receptor (VEGFR), and platelet-derived growth factor receptor (PDGFR), can stimulate the activity of PI3K/AKT and MAPK/ERK. When the AKT pathway is inhibited, it sends out signals that can activate transcription factors, including transcription factor EB (TFEB), nuclear factor-kappa B (NF-кB), members of the forkhead box class O transcription factors (FOXO), and p53, to regulate the expression of autophagy-related genes [26]. Class-III PI3K (PI3KC3) complexes, such as PI3K3/VPS34 and p150/VPS15, can regulate vesicular nucleation. Phagophore membrane assembly can be activated by the ULK1 complex. Positive regulators (ATG14L, Bif-1/UVRAG, and Ambra-1) can interact with the PI3KC3 complex composed of proautophagic proteins that can bind to microtubules. Following ULK1 phosphorylation, Ambra-1 and the PI3KC3 complex are released from microtubules, forcing the PI3KC3 complex to move to the ER for autophagosome formation [44,45,46,47]. 

Many proteins involved in autophagosome maturation and elongation can be recruited by Beclin-1 [32]. ATG12–ATG5–ATG16L and phosphatidyl-ethanolamine (PE)-light chain 3 (LC3) are two ubiquitin-like conjugation systems involved in phagophore elongation and autophagosome formation [48,49]. In the ATG12–ATG5–ATG16L system, E1/E2 ligases ATG7 and ATG10 can initiate the binding of ATG5 and ATG12 to form the ATG5–ATG12 complex. On the outside of the extended autophagosomal membrane, the ATG5-ATG12 complex can oligomerize and generate a larger ATG16L system. The ATG16L complex can detach from the autophagosomal membrane during autophagosome development [48,50]. Phagophore elongation in the second system is mediated by LC3, which has been changed to LC3-I. During the process, ATG7 and ATG3 can catalyze the formation of LC3-II-PE (MAP1LC3B) by attaching LC3-I to the E1 and E2 ligases of PE. The LC3-II-PE complex can bind to the inner and outer surfaces of the autophagosome, aiding the attachment of p62 adaptor proteins, allowing it to absorb and degrade substrates [21,35,51]. The autophagosome membrane is also extended and closed by LC3-II [52].

Mature autophagosomes may form amphisomes by fusing with endocytic vesicles or directly fusing with lysosomes, where the material is degraded by several lysosomal hydrolases and macromolecule precursors, which are recovered or used to fuel metabolic pathways [38,41]. The autophagosome’s internal membrane is degraded in an LC3-dependent manner [13]. The fusion of the autophagosome and lysosome is enabled by an autophagosomal soluble N-ethylmaleimide-sensitive factor attachment protein receptor (SNARE) protein - Syntaxin 17 (Stx17), a small G-protein RAB7, and a membrane tethering homotypic fusion and vacuole protein sorting (HOPS) complex [50]. UVRAG also interacts with C vacuolar protein (C-VPS) on the autophagosome, which promotes Rab7-GTPase activity in conjugation with LAMP-1/2, resulting in autophagosome–lysosome fusion. As a result of autophagosome–lysosome fusion, the inner membrane is formed and cytosolic materials are degraded by lysosomal hydrolases [14,26,53]. The degradation of intravesicular products and autophagic charge by acid hydrolyses in the lysosome is the final step in the autophagic process. Degrading products are recycled and returned to the cytosol [54]. Lysosomal proteases, mainly cathepsins, can degrade the cargo inside the lysosome and release biomolecules in the cytosol for reuse [55].

## 3. Autophagy Regulation

### 3.1. AMPK/mTORC1/ULK1 Pathway

The central regulator for the activity of ULK1 and mTORC1 is AMPK, a Ser/Thr kinase that can monitor changes and intracellular energy levels to balance metabolic flow [56,57]. AMPK is inactive while mTORC1 is active when glucose levels are adequate. ULK1 is negatively regulated by active mTORC1 by direct phosphorylation at Ser757, which prevents ULK1 from interacting with and activating AMPK [42,56]. Ubiquitination of ULK1 is mediated by phosphorylation of AMBRA1 and the impairment of TRAF6 [56]. By phosphorylating TSC2 and Raptor in the ULK1 autophagic complex, AMPK is stimulated while mTORC1 is inhibited as cellular energy level decreases [57]. Ser757 phosphorylation is then decreased and ULK1 can interact with AMPK on Ser317 and Ser777. The phosphorylation of ULK1 by AMPK becomes active and can initiate autophagy. The phosphorylation of ULK1 by AMPK thereby initiates autophagy [42]. ULK1 can also trigger autophagy by phosphorylating Ambra-1 and BECN1. Such phosphorylation then activates VPS34 and causes autophagosome maturation from the ER. The autophagolysosome is formed during the fusion of the autophagosome and lysosome, resulting in the breakdown of internal contents that can be recycled [56]. The AMPK/mTORC1/ULK1 signaling pathway can also regulate the PIK3C3/VPS34 complex by activating or inhibiting Beclin-1 and VPS34, depending on the presence of an autophagy-specific subunit ATG14L. When ATG14L is present, Beclin-1 is phosphorylated by the AMPK to activate the complex [58].

### 3.2. PI3K/AKT/mTOR Pathway

Unlike other signaling pathways, the PI3K/AKT/mTOR pathway is associated with a variety of human disorders, including neurodegenerative diseases, ischaemic brain injury, and tumors. In cancers, the PI3K/AKT/mTOR signaling pathway is involved in tumor cell growth, proliferation, invasion, and metastasis; endothelial cell growth; and angiogenesis, radiation sensitivity, apoptosis, and autophagy inhibition [59,60,61,62]. Class I PI3K can respond to growth factor signaling by producing phosphatidylinositol (3,4,5)-trisphosphate (PtdIns(3,4,5)P3), which inhibits autophagy through the AKT-mTORC1-ULK1 complex, whereas class III PtdIns3K, particularly ATG14-containing PtdIns3K-C1 and class II PI3K, can contribute to autophagy initiation and progression by producing PtdIns3P. The autophagosome formation is positively regulated by PtdIns3P. When growth factor levels are low, the PIK3CB subunit of class I PI3K can regulate PI3K-C1 through Rab5A, thus inducing autophagy. PIK3C3 can also act as an amino acid sensor and a mTORC1 activator [63].

### 3.3. MEK/ERK1/2 Pathway

Of all MAPK signaling pathways for the growth and survival of tumor cells, the Ras/Raf/MAPK (MEK)/ERK signaling pathway is the most important one. MAPK cascades control a variety of functions, including cellular stress response, proliferation, differentiation, and apoptosis [64], while the ERK (ERK1/2) signaling pathway controls autophagy regulation [65]. MEK/ERK can maintain activities of mTORC1/mTORC2 at the basal levels and prevent them from being disassembled. This prevents Beclin-1 from initiating an autophagic response. However, upon autophagy stimulation, upstream regulator AMPK can activate MEK/ERK, which can increase the Beclin-1 level by disassembling mTORC1 or mTORC2. As a result, the Beclin-1 level is significantly elevated, causing cytodestructive autophagy, whereas a sustained or moderately elevated Beclin-1 level can induce cytoprotective autophagy. Although ERK regulates autophagy, MEK can bypass ERK to activate autophagy [66]. In cancer cells, the interplay between MAPK and AMPK signaling can alter the autophagic process. In Ras/Raf-mutated cancer cells, MAPK modulation by RAF/MEK/ERK inhibitors can promote autophagic flux, which maintains cellular metabolic hemostasis and contributes to MAPK inhibition tolerance [67].

### 3.4. Transcriptional Regulators of Autophagy

#### 3.4.1. Transcription Factor EB (TFEB)

TFEB is a transcriptional regulator that promotes the expression of genes associated with autophagosome formation, lysosomal function, and biogenesis [68]. mTOR phosphorylation at the cytoplasm can regulate TFEB’s subcellular localization and activity [69]. TFEB’s nuclear localization and activity are regulated by ERK2, which is modulated by the abundance of extracellular nutrients, while the MAPK-dependent mechanism controls the biogenesis and interaction of these two distinct cellular organelles [70]. Once TFEB is localized from the cytosol into the nucleus, it can activate the expression of various autophagy-related molecules and induce the autophagic process [71]. These autophagy-related molecules include BECN1, WIPI1, ATG9B, and NRBF2 for autophagy initiation; GABARAP, MAP1LC3B, and ATG5 for autophagosome membrane association; SQSTM1 for substrate capture; and UVRAG and RAB7 for autophagosome trafficking and fusion with lysosomes [72]. When TFEB is combined with the E-box element of coordinated lysosomal expression and regulation (CLEAR), it can activate LAMP1 expression and promote lysosomal formation [71].

#### 3.4.2. Forkhead Box O (FOXO)

The FOXO family transcription factors are important regulators of cellular homeostasis, aging, stem cells, angiogenesis, autophagy, and signaling pathways for cancer growth and metastasis [73,74]. In mammals, the FOXO family consists of four members: FOXO1/FKHR, FOXO3/FKHRL1, FOXO4/AFX, and FOXO6 [75,76]. The activity of FOXO is influenced by external energy changes, growth factor activation, and nutritional level [74]. Post-translational modifications (PMTs), such as AMPK-induced phosphorylation and PRMT6-induced methylation, can aid FOXO translocation from the cytosol to the nucleus, which then enables the expression of autophagy-related molecules, such as ULK1/2, Beclin-1, and ATG14 for autophagy induction; MAP1LC3B, GABARAPL, and ATG4 for elongation; and TFEB and Rab7 for autophagosome–lysosome fusion. It also increases the expression of Sestrin 3 (Sesn3), which can activate AMPK and deactivate mTORC1, thus activating autophagy through the mTORC1/ULK1 pathway, although it does not directly trigger autophagy [77].

#### 3.4.3. Tumor Protein p53

p53 is a transcription factor that controls important tumor suppressor pathways, including senescence, cell cycle arrest, apoptosis, and DNA repair, to maintain a cell’s genomic integrity and to prevent cell proliferation with unrepaired damaged DNA [78]. Stress-induced cell cycle response is related to p53 and autophagy regulation [79]. When p53 is exposed to oncogenic, genotoxic, or hypoxic stress, it undergoes reversible PTMs that enable it to be activated [80]. Its activation then induces the expression of autophagy-related molecules such as ULK1, ATG2, ATG4, ATG5, ATG7, ATG10, and GABARAP [79,81]. The p53 can enhance autophagy by inducing distinct target genes, while cytoplasmic p53 inhibits autophagy. Autophagy can be enhanced by alternative reading frame protein (ARF), a p53 upstream regulator. ATG7, a key autophagy regulator, can activate p53 and induce p21, which helps cells survive under a nutrient-deprived condition [82].

#### 3.4.4. Hypoxia-Inducible Factor-1 (HIF-1)

HIF-1 is an important transcription factor that enables cells to adapt and function in hypoxic environments. HIF-1 is normally destroyed by proteosomes. However, it is significantly upregulated in hypoxia to activate several cellular functions, including angiogenesis, erythropoiesis, energy metabolism, pH modulation, cell proliferation, and tumor growth [83,84]. Hypoxia can stabilize HIF-1α from the cytoplasm to the nucleus. Stabilized HIF-1α can interact with the HIF-1β subunit to form HIF-1. As a result, the HIF-1α/HIF-1β complex becomes transcriptionally activated through a range of coactivators, causing autophagy, survival, cell proliferation, and regulation of angiogenesis [85]. Phosphoglycerate kinase 1 (PGK1), glucose transporters 1/3 (GLUT1/3), lactate dehydrogenase (LDHA), enolase 1 (ENO1), hexokinases (HK1/2), pyruvate dehydrogenase kinase 1 (PDK1), and 6-Phosphofructokinase-2 (PFK-2) or fructose bisphosphatase-2 (FBPase-2) are all influenced by HIF-1 [86].

## 4. Paradoxical Roles of Autophagy in GBM

Autophagy is a highly conserved and controlled mechanism that can direct the lysosomal degradation of proteins and damaged organelles to maintain genomic stability, metabolism, and cell survival [87]. Previous studies have shown that the involvement of autophagy in cancer tumorigenesis is complex and context-dependent [41]. Autophagy can have a dual function in GBM (Figure 3). It can destroy harmful unfolded proteins, oncogenic protein substrates, and damaged organelles as a tumor suppressor (Table 2). It may also promote tumor development by recycling metabolic substrates and preserving the mitochondrial functional pool (Table 3) [88].

### 4.1. GBM Is Suppressed by Autophagy

GBM can be suppressed by autophagy through autophagic cell death. In U343 glioma cells, adenovirus-mediated shRNA inhibiting c-Met can induce autophagy that is characterized by many vacuolated cells, upregulated Beclin-1 expression, and recruitment of MAP1LC3. The autophagy induction inhibits cell proliferation and induces cell cycle arrest at the G2/M stage, leading to autophagic cell death. In vivo, adenovirus-mediated shRNA intratumoral injection in male athymic nu/nu mice can reduce tumor growth and increase overall survival [89]. Likewise, when Beclin-1 is overexpressed in U87MG cells, it can increase cellular autophagy and cause apoptosis by binding to Bcl-2 and Bcl-xL, releasing cytochrome c into the cytosol and activating caspases-3/-9 [90]. Hence, decreased expression levels of Beclin-1 and LC3B-II can lead to astrocytic tumor progression [91]. Moreover, knockdown of Beclin-1, VPS34, and ATG5 can inhibit autophagy in IR-induced cell death in U87MG, U373MG, and LN229 glioma cells [92]. 

Meanwhile, inhibiting the STAT3/Bcl-2/Beclin-1 signaling pathway by upregulating miR-519a can sensitize the effect of TMZ and cause autophagy-mediated apoptosis in U87MG cells in vitro and a xenograft model in vivo [93]. On the other hand, overexpression of miR224-3p can suppress hypoxia-induced autophagy, while knocking down endogenous miR224-3p can boost autophagic development in normoxia. miR224-3p overexpression can attenuate the proliferation and promote hypoxia-induced apoptosis of U87MG and U215 glioma cells in vitro and reduce tumor volume in athymic nude mice in vivo [94]. 

ULK2-induced autophagy can also inhibit cell growth and proliferation of LN229 glioma cells. The growth inhibition by ULK2 involves catalase degradation and ROS generation. Moreover, apoptosis plays a major role in ULK2-induced cell death, especially when high levels of autophagy are induced. In contrast, downregulation of ULK1/2 can inhibit autophagy and promote astrocyte transformation [95]. Prolonged hypoxia can induce autophagic cell death in U87MG and U373MG through a BCL2 interacting protein 3 (BNIP3)-dependent mechanism [96]. Similarly, cellular autophagy can be induced by co-treatment of TMZ and carnosic acid in U251 and LN229 glioma cells by AKT phosphorylation inhibition, p62 downregulation, and LC3-I to LC3-II transition. Cyclin B1 inhibition and PARP and Caspase-3 activation can lead to cell cycle arrest and cellular apoptosis [97]. In addition, interfering with EGFR and subsequent IR and TMZ treatment can induce the autophagic cell death of T98G and U373MG cells. Autophagy suppression can reduce the effect of combined IR and EGFR interference, while rapamycin-mediated autophagy activation can increase the effect of combined EGFR/IR treatment [98].

**Table 2 cells-10-01411-t002:** Autophagy regulation suppresses gliomagenesis.

Autophagy Regulation	Result	Glioma Model	Ref
Beclin-1 overexpression	cell proliferation inhibition, cell cycle arrest at the G2/M; apoptosis	U343, U87MG	[89,90]
STAT3/Bcl-2 downregulation; Beclin-1 overexpression	apoptosis	U87MG	[93]
miR224-3p overexpression	hypoxia-induced apoptosis; reduced tumor volume	U87MG, U215, athymic nude mice	[94]
ULK2 overexpression	autophagic cell death; cell growth and astrocyte transformation inhibition	LN299; NIH nu/nu nude mice	[95]
BCL2 Interacting Protein 3 activation	hypoxia-induced cell death	U87MG, U373	[96]
Cyclin B1 downregulation/PARP and Caspase-3 upregulation	cell cycle arrest, apoptosis	U251, LN229	[97]
AKT/mTOR pathway inhibition	cell growth inhibition; reduced tumor volume	U87MG; A172 nu/nu nude mice	[99,100,101]
EMT regulators downregulation/N- and R-cadherin upregulation	inhibits migration and invasion capacity	GL15, U87MG	[102]
EGFR modulation	inhibits clonogenic and migration	IRT98G, U373MG	[98]
H3L9 demethylation	suppresses cell proliferation	LN229, U87MG	[103]
AKT phosphorylation	senescence; reduces tumor growth	U343 athymic nu/nu mice	[89]
MAPK14/p38 and PRKAA/AMPK/ULK1 pathway activation	senescence	U87MG	[104]

Autophagic induction could also impair cell growth and proliferation. Rapamycin, an mTOR inhibitor, can inhibit cell growth by inducing autophagy in U87MG cells and primary cell cultures derived from various GBM patients. Rapamycin can nearly double the survival time of mice when it is given to brain xenografts in nude mice in vivo and inhibits tumor volume by more than 95% [99]. On the other hand, ULK2 overexpression can suppress LN299 cell growth and proliferation, inhibit astrocyte transformation in vitro, and impair tumor development in NIH nu/nu nude mice in vivo [95]. Itraconazole can induce autophagy by disrupting cellular cholesterol redistribution. It can also suppress the AKT/mTOR signaling pathway, resulting in U87MG cell proliferation inhibition [100]. 

On the other hand, autophagy can inhibit GBM migration and invasion. GBM migration and chemokine-mediated invasion are both inhibited by autophagic stress induced by starvation or pharmacological inhibitors. This will lead to the downregulation of EMT regulators and the upregulation of N- and R-cadherins, resulting in the impaired migration and invasion capacity of GL15 and U87MG cells. In contrast, when Beclin-1 is knocked down, GBM cell migration capacity is increased along with the upregulation of SNAIL and SLUG, while N- and R-cadherin mRNA expression levels are decreased. Downregulation of ATG5 and ATG7 and upregulation of EMT regulators could also increase GBM cell migration and invasion [102]. Similarly, EGFR modulation by rapamycin-mediated autophagy induction can cause IR-T98G and in U373MG cells to lose their clonogenic and migration capacities [98]. TGF-b signaling downregulation and increased autophagic activity can also inhibit the adhesion and migration of U87MG cells [105]. Moreover, downregulation of the PI3K/AKT/mTOR signaling pathway can reduce the migration and invasion of A172 glioma cells [101]. In T98G cells, autophagy can modulate cell cycle arrest regulated by the PI3K/AKT/mTOR signaling pathway and migration inhibition mediated by the suppression of MMP-2/-9 activity involved in the MAPK signaling pathway [106]. G9a deficiency, a lysine methyltransferase that can mono- and dimethylate histone H3 lysine 9 (H3L9), can suppress cell proliferation, while overexpression of full-length c-Myc can restore autophagy activation in LN-229 and U87MG glioma cells [103].

Autophagic induction can cause senescence in GBM cells. For example, shMet expressing adenovirus can suppress AKT phosphorylation and increase the number of senescence-related genes (PAI-1, TGase II, and SM22) in U343 glioma cells [89]. Moreover, sustained modulation of AKT/mTOR pathway and transient activation of MAPK14/p38 and PRKAA/AMPK/ULK1 pathways by TMZ treatment can induce autophagy, leading to senescence and causing DNA damage in U87MG cells [104]. TMZ treatment in LN-229 and U87 MG glioma cells can undergo autophagy, senescence, and apoptosis in a specific time-dependent manner. Cellular senescence increases with postexposure time and, similar to autophagy, precedes apoptosis. If autophagy is suppressed, TMZ-induced senescence is reduced. All of these effects are completely abolished in MGMT-expressing isogenic glioma cells, suggesting that a single form of O^6^MeG DNA lesion is capable of triggering all of these responses [107].

### 4.2. GBM Is Promoted by Autophagy

Several autophagy-related proteins are overexpressed during gliomagenesis. Autophagy proteins are expressed more often in high-grade gliomas than in low-grade gliomas. An increased autophagic rate or a blockage at autophagy’s final degradation stage may explain the increased level of autophagic proteins in high-grade gliomas [108]. Autophagy is also increased in astrocytomas compared with normal CNS tissues, although such an increase is unrelated to the WHO grade or patient survival [109]. Meanwhile, ATG proteins including ULK1, ULK2, LC3A, LC3B, Beclin-1, and p62 have varying degrees of cytoplasmic overexpression in GBM. Cathepsin D, LAMP2a, and TFEB levels are also elevated in GBM. TFEB is linked to PTEN, Cathepsin D, HIF-1α, LC3B, Beclin-1, and p62 expression, while PTEN is linked to LC3B expression [110]. ATG proteins, such as LC3B, p62, CTSB, and LAMP2, are also upregulated in perinecrotic areas in GBM, implying that changes in the microenvironment can promote autophagy [109]. Inhibition of autophagy can significantly reduce the development of GBM, demonstrating its importance in tumor formation. ULK1, ATG7, and ATG13 knocked down by shRNA can result in autophagy inhibition as measured by endogenous LC3 lipidation in an RCAS/TVA mouse model, although residual maintenance of autophagic activity is observed [111].

**Table 3 cells-10-01411-t003:** Autophagy regulation promotes gliomagenesis.

Autophagy Regulation	Result	Glioma Model	Ref
Malat1 activation	promotes cell proliferation	U87MG, U118, U251, U373, D247	[112]
ATG4C activation	promotes cell proliferation, prevents apoptotic cell death, resistance to treatment	U87MG	[113]
TRIM28 activation	promotes cell proliferation	U251	[114]
Pentraxin 3 activation	promotes GBM progression	U87MG	[115]
Bax protein downregulation	prevents apoptotic cell death	U87MG	[116]
BNIP3L	aids tumor progression	-	[117]
ATG9A activation	promotes GBM growth and proliferation	-	[118]
ATG5 activation	promotes vasculogenic mimicry	GSC primary culture	[18]
mTOR activation	promotes vasculogenic mimicry	U87MG	[35]
Beclin-1, HIF-1, VEGF, and MMP2 expression	increase in migration rate, the total length of VM tubes, invasion cell number	U87MG	[119]
TGF-1 activation	promotes EMT and tumor progression	C6 glioma cells	[120]
TGF-2 activation	promotes GBM progression and cell invasion	U251, T98, U87MG	[121]
TMZ induced autophagy	enhances cell migration, EMT infiltration; GBM resistance to treatment	U87MG	[122,123]
TRPC5 activation	GBM resistance to treatment	U87MG	[124]
Autophagy activation	GBM resistance to treatment	U87MG; U251	[125,126,127]
MDA-9 activation	GSC sensitivity to anoikis	-	[127]
ATF4 activation	resistance to anoikis and metastasis tumor cells	-	[128]
mTORC1/mTORC2 activation	self-renewal	-	[129,130]
EGFRvIII/CD133 coexpression	promotes stemness	-	[131]
HIFs expression	promotes self-renewal, proliferation, and survival	-	[132]
WNT-CTNN1B signaling pathway activation	aids cancer cell proliferation and stemness	-	[133]

Autophagy also contributes to GBM proliferation. In U87MG, U118, U251, U373, and D247 glioma cells, autophagy activation and cell proliferation can be triggered by metastasis-associated lung adenocarcinoma transcript 1 (Malat1). Malat1-induced cell proliferation can be suppressed by inhibiting autophagy with 3-MA. Malat1 induces autophagy and promotes cell proliferation by sponging miR-101 and upregulating STMN1, RAB5A, and ATG4D expression in glioma [112]. ATG4C, an autophagy regulator involved in pro-LC3 cleavage and LC3 II delipidation, can aid gliomagenesis. When ATG4C is knocked out, the proliferation of U87MG glioma cells is inhibited by causing cell cycle arrest in the G1 process. Furthermore, BALB/C male nude mice with ATG4C knockdown have a significant reduction in glioma development [113]. The autophagy mechanism is also aided by the tripartite motif-containing protein 28 (TRIM28). Levels of TRIM28 and autophagy are increased dramatically as tumor grade increases, thus promoting the proliferation of U251 glioma cells [114]. Furthermore, pentraxin 3, a GBM prognostic biomarker, can promote GBM progression by inhibiting tumor cell autophagy in U87MG glioma cells [115].

Autophagy-related proteins can prevent apoptotic cell death in GBM. As 3-MA can inhibit autophagy in U87MG cells, it can cause upregulation of the pro-apoptotic protein Bax, which activates melatonin-induced cell apoptosis [116]. Meanwhile, when ATG4C is depleted in U87MG cells, it can trigger apoptosis through ROS accumulation [113]. On the other hand, hypoxia-induced autophagy contributes to GBM survival. Both oncogenic transformation of KRAS-driven glial cells and sustained viability under stress require autophagy, which keeps growth signaling pathways active in cells deprived of growth factors under hypoxia [111]. Likewise, hypoxia-induced autophagy via BNIP3 and BNIP3L is a survival mechanism that can aid tumor progression [117]. Similarly, hypoxia can reliably induce ATG9A. Silencing ATG9A will result in reduced GBM proliferation and delayed tumor progression in vitro. ATG9A is needed for GBM growth in both normoxic and hypoxic environments. It can regulate autophagy activation during hypoxia. In orthotopic patient-derived GBM xenograft models, inhibiting ATG9A expression can effectively stop tumor development [118].

Meanwhile, by inducing autophagy, astrocyte elevated gene 1 (AEG-1) can improve malignant glioma cell resistance to transforming growth factor-beta 1 (TGF-β1)-induced epithelial-mesenchymal transition (EMT). In malignant glioma cells, autophagy is induced by TGF-β1, which activates AEG-1 via phosphorylation of Smad2/3. As a result, increases of oncogene cyclin D1 and EMT markers can lead to tumor progression. Administration of siRNA-AEG-1 to rats implanted with C6 glioma cells in vivo can inhibit tumor growth and increase the incidence of apoptosis of tumor cells [120]. Transforming growth factor-beta 2 (TGF-β2)-induced autophagy is also essential for glioma invasion because it affects epithelial-mesenchymal transformation and metabolism conversion, with particular effects on mitochondria trafficking and membrane potential. In U251, T98, and U87MG cells, autophagy can induce feedback on TGF-β2 by maintaining its autocrine loop and influencing the expression of Smad2/3/7. TGF-β2 can promote glioma cell invasion by inducing autophagy through Smad and non-Smad pathways. Moreover, an in vivo experiment using a NOD-SCID mouse model has revealed the advantages of using TGF-β and autophagy pathway inhibitors together, as CQ could enhance the effect of TGF-β inhibitors. This result adds to the growing body of evidence that TGF-β plays a role in glioma invasion in part by autophagy [121]. On the other hand, induction of autophagy by TMZ also enhances cell migration and infiltration via the EMT in U87 cells harboring wild-type (WT) TP53, but not in U251 cells harboring mutant (MU) TP53 cells. The knockdown of ATG7 and BECN1 could inhibit autophagy and interrupt the effects of TMZ in glioma cells [122].

Resistance of GBM to TMZ is promoted by the transient receptor potential cation channel subfamily C member 5 (TRPC5) via the CAMMKβ/AMPKα/mTOR pathway. TMZ-resistant U87MG cells retain a high basal autophagy level, while silencing TRPC5 expression or inhibition of autophagy can reverse TMZ resistance. Downregulation of TRPC5 to inhibit autophagy can increase TMZ sensitivity in shRNA-TRPC5 treated U87MG xenografts in vivo [124]. ATG4C deficiency can also suppress TMZ-activated autophagy and increase U87MGcell sensitivity to TMZ [113]. Autophagy can protect U87MG cells from ZD6474’s proapoptotic effects, leading to tumor sensitivity to the treatment. The knockdown of both ATG7 and Beclin-1 genes and the use of chloroquine and 3-MA therapy can inhibit ZD6474-induced autophagy in U251 and U87MG cells in vitro and in female BALB/c nude mice in vivo. The increased sensitivity of GBM cells to ZD6474-induced apoptosis can result in decreased cell viability in vitro [125]. GBM is also resistant to Bevacizumab, an antiangiogenic treatment. Hypoxia-mediated autophagy can facilitate the survival of U87MG and T98G cells in vitro. Thus, the knockdown of ATG7 can promote bevacizumab responsiveness in a U87MG/shATG7 xenograft model **[134]**. Furthermore, autophagy inhibition has been observed in 3-MA or ATG5 siRNA-treated U251 glioma cells. STAT3 inhibition can lead to increased radiosensitizing activity. As a result of the simultaneous autophagy inhibition and STAT3 signaling, more apoptotic cells are induced [126]. In LN18 and LN229 cells, the Wnt/β-catenin signaling pathway regulates ATG9B, a crucial downstream component of TMZ-induced autophagy. As a result, using a Wnt-inhibitor in TMZ-resistant GBM can enhance the effect of conventional TMZ therapy [123].

Glioma stem cells (GSCs) are a subpopulation of tumor cells with stem-like properties that can self-renew and give rise to heterogeneous cells that make up a tumor. They are thought to be responsible for tumorigenesis, maintenance, and recurrence [135]. Inhibiting mTORC2, but not mTORC1, has a significant effect on GBM cell proliferation and metabolic activity, indicating that suppressing mTORC2 may be a useful tool for managing GBM growth and counteracting GSC proliferation, which has been linked to GBM relapse [129]. Meanwhile, GSCs can form vasculogenic mimicry (VM), a microvascular circulation that is independent of VEGF. In NOD-SCID mice, inhibiting angiogenesis and autophagy can slow the development of GSC-formed tumors and VM. Autophagy in GSCs can promote the development of VM in a microenvironment with poor blood supply, providing the nutrients needed for tumor development. Bevacizumab-induced macroautophagy in GSCs has been linked to tumor resistance to antiangiogenic therapy and the development of VM. Chloroquine or the knockdown of ATG5 can prevent VM development and kinase insert domain receptor (KDR) phosphorylation in GSCs, while rapamycin can enhance the number of VM and boost the phosphorylation of KDR/VEGFR-2 [18]. Similarly, VM structures have been linked to mTOR expression in glioma cells. Under both normoxia and hypoxia conditions, rapamycin, the mTOR inhibitor, can suppress VM formation in U87MG cells. HIF-1 and other molecules involved in the VM formation signaling cascade can be inhibited by rapamycin and mTOR siRNA [136]. Meanwhile, under hypoxic conditions, migration rate, the total length of VM tubes, invasion cell number, and expression of Beclin-1, HIF-1, VEGF, and MMP2 are significantly increased in U87MG cells. Hence, silencing Beclin-1 can significantly reduce hypoxia-induced VM development, invasive and migratory abilities, and expression levels of VM-related molecules, while it can significantly reduce autophagic flux [119].

Expressing both CD133 and EGFRvIII in GBMs can lead to the stemness of cancer stem cells. Furthermore, CD133+/EGFRvIII+/EGFR- cells may initiate tumor formation and play a role in gefitinib resistance [131]. In vitro, inhibiting HIFs in GSCs can decrease self-renewal, proliferation, survival, and GSC tumor initiation potential in athymic BALB/c nu/nu mice. HIF-1 is needed for the induction of VEGF expression in both glioblastoma stem cells and non-stem cells by transcriptionally controlling the VEGF promoter, while HIF2 is required for VEGF expression in GSCs [132]. Cancer cell proliferation and stemness are aided by WNT-CTNN1B signaling. TCF inhibition or silencing of TCF4 or CTNNB1/β-catenin can upregulate SQSTM1/p62 transcription and protein levels in GBM, resulting in increased autophagy. Autophagic flux can also be induced by DKK1/Dickkopf1, a canonical WNT receptor antagonist. TCF inhibition regulates autophagy through MTOR inhibition and dephosphorylation as well as TFEB nuclear translocation. As a result, TCF inhibition or silencing has an impact on GBM cell proliferation and migration [133]. 

Finally, the sensitivity of GSC to anoikis, a form of programmed cell death that occurs when anchorage-dependent cells separate from the extracellular matrix (ECM), is boosted by protective autophagy. GSCs display defensive autophagy and anoikis resistance in nonadherent settings, which are linked to melanoma differentiation-associated gene-9/Syntenin (MDA-9) expression. In vivo, inhibiting MDA-9 in GSCs can cause anoikis and cell death by reversing protective autophagy [127]. Activating transcription factor 4 (ATF4), an ISR’s master transcriptional effector, can protect transformed cells from anoikis. ATF4 can initiate a coordinated program of cytoprotective autophagy and antioxidant reactions in response to the loss of attachment, including the induction of the antioxidant enzyme heme oxygenase 1 (HO-1). HO-1 is involved in ATF4-dependent resistance to anoikis and metastasis of tumor cells [128].

## 5. Autophagy as a Therapeutic Target for GBM

While TMZ has been widely used to treat primary and recurrent high-grade gliomas, tumor recurrence and development limit their effectiveness. Recent research has revealed that TMZ treatment can cause autophagy [137]. Autophagy is noted for its function in cellular homeostasis. It is elevated in response to stressors such as nutrient and oxygen depletion as well as anticancer treatment [43]. However, its functions in cancer growth and treatment response are debatable [19]. To interfere with the survival mechanism of GBM, various treatment modalities have been established to induce or inhibit autophagy. These innovative therapies complement existing standard GBM treatments such as TMZ and IR.

### 5.1. Autophagy Inhibitors

#### 5.1.1. 3-Methyladenine (3-MA)

3-Methyladenine (3-MA) is commonly used as an inhibitor of the PI3KC3 [138]. Although 3-MA can suppress starvation-induced autophagy, when it is used for a long time, it promotes autophagy flux in a nutrient-rich environment. The stimulatory function of 3-MA is determined by differences in its temporal effects on PI3KC1 and PI3KC3. Although 3-MA can persistently inhibit PI3KC1, it can only transiently inhibit PI3KC3 [139]. Some studies have shown that 3-MA can effectively cure autoimmune neuritis, control enterovirus infection, and pathogenesis, and suppress tumor metastasis in an autophagy-independent way [138]. In GBM, a combination of 3-MA and cisplatin, a chemotherapeutic drug, can increase cisplatin-induced apoptosis in U251 human glioma cells by increasing ER tension. This suggests that inhibiting autophagy may increase the efficacy of cisplatin chemotherapy [140]. When 3-MA is combined with melatonin, it can lower Bcl-2 expression but increase Bax expression. As a result, 3-MA can disrupt autophagy, allowing melatonin to cause apoptosis of A172 and U87MG cells [116].

#### 5.1.2. Chloroquine (CQ)

Chloroquine (CQ), a 4-aminoquinoline lysosomotropic drug, is often used to treat rheumatoid arthritis, malaria, sarcoidosis, liver amoebiasis, and lupus erythematodes. CQ can build up in the lysosome, raising the intralysosomal pH and preventing autophagosome–lysosome fusion, which is needed for autophagy [141]. It has been shown that CQ can improve the antiproliferative effect of radiation on cancer cells, likely by negatively regulating DNA repair processes and removing chemo-resistant cancer cells. CQ can also significantly increase the overall survival time of GBM patients when it is paired with conventional GBM therapies [142]. CQ can cause autophagic cell death in LN308 and U87MG cells in a dose-dependent manner by accumulating autophagic vacuoles, affecting cathepsin D levels and subcellular distribution, thus altering the function of the lysosome and resulting in cell death [142]. In highly radioresistant GSCs, CQ can strongly promote IR-induced cell death. Triple combinations of PI3K/AKT inhibitor, CQ, and IR can decrease the dosage of CQ needed to cause cell death [143]. Furthermore, combining CQ and IR treatment can increase U87MG cell apoptosis by increasing caspase-3 levels, decreasing Bcl-2 levels, and inducing cell cycle arrest at the G1/G0 stage [144]. TMZ and CQ cotreatment can increase mitochondrial ROS and promote cell death in U87MG cells [145]. Similarly, combining TMZ and CQ treatment in U87MG cells can decrease cell proliferation by increasing apoptosis [146]. Based on preclinical and early clinical findings, CQ was designated as an orphan drug in the EU in November 2014 and the US in May 2015 for the treatment of glioma [141]. GBM patients can benefit from CQ and chemoradiation cotreatment, which inhibits autophagy, thus making tumor treatment more effective according to preclinical findings [141]. Clinical trials have shown that a daily dosage of CQ at 200 mg is the best-tolerated dose when paired with IR and reciprocal TMZ in newly diagnosed GBM patients [141].

#### 5.1.3. Quinacrine

Quinacrine (QC), an antimalarial drug, can significantly increase TMZ cytotoxicity in both resistant and sensitive cells. Changes in expression levels of LC3II, p62, and cleaved caspase-3 and enhanced apoptosis may accompany such cytotoxic effects of QC. This indicates that QC could at least partially sensitize GBM cells to TMZ by inducing apoptosis [147]. Hypoxia, on the other hand, can enhance the efficacy of cediranib/QC by reducing cell viability and causing a significant increase in apoptosis of 4C8 glioma cells. In response to cediranib, QC, and hypoxia, levels of LC3-II, an autophagic vacuole biomarker, are increased significantly. In GBM cells, a combination of cediranib and QC can inhibit AKT phosphorylation [148]. Meanwhile, SI113, a small molecule SGK1 inhibitor, combined with quinacrine can synergistically inhibit GBM growth properties in ADF, U373MG, and T98G GBM cells, including neurospheres [149].

#### 5.1.4. ABT-737

By neutralizing Bcl-2 family proteins, ABT-737 can inhibit antiapoptotic pathways and cause the release of Bax and Bak proteins, thus inducing apoptosis [150]. ABT-737 has raised new possibilities for a potential inhibitor-based drug Bcl-2 family protein that could be used as a single agent treatment for a variety of cancers, including GBM [151]. By releasing Bax proapoptotic protein from its binding partner Bcl-2, ABT-737 therapy can effectively stimulate apoptotic cell death in U87MG and U251MG cells. It has been shown that local administration of ABT-737 can improve the survival of an intracranial glioma xenograft model [152]. ABT-737 can also inhibit Bcl-2L12’s antiapoptotic function and make U87MG and T98G GBM cells more responsive to TMZ [151]. By targeting Bcl-2, Bcl-xl, and Bcl-w antiapoptotic proteins, ABT-737 can kill both neuroblastoma SH-SY5Y cells and GBM T98G cells [153].

#### 5.1.5. Obatoclax

Obatoclax (GX 15–070), a small molecule inhibitor, can prevent Bak/Mcl-1 binding and stimulate cytochrome C release and caspase-3 activity by modulating Bim. Clinical trials of obatoclax for hematological tumors, non-small-cell lung carcinoma, and advanced-stage small-cell lung carcinoma are currently in progress [154]. Meanwhile, elevated levels of the autophagy marker LC3-II indicate that obatoclax can promote apoptosis and autophagy in neuroblastoma cells. When autophagy is inhibited by hydroxychloroquine (HCQ), the cytotoxicity of this combination treatment of HCQ and obatoclax is increased, implying that autophagy induced by these agents has a cytoprotective effect. An in vivo experiment has shown that a combination of obatoclax and HCQ can significantly decrease tumor growth and the number of distant metastases [155].

#### 5.1.6. Bortezomib

Bortezomib (BTZ), a potent proteasome inhibitor, can inhibit the NF-kB pathway. It has a chemo-sensitizing effect when combined with other antitumoral drugs [156]. Pretreatment with BTZ before TMZ can destroy chemo-resistant GBM cells in vitro by depleting MGMT mRNA and protein without affecting the methylation of the MGMT promoter. Furthermore, BTZ can cross the blood-brain barrier (BBB) with reduced proteasome activity, allowing animals to live longer [157]. On the other hand, BTZ can inhibit U87MG and U251 cell proliferation by triggering cell cycle arrest and apoptosis in a time- and dose-dependent manner. BTZ can also inhibit cell colony formation, spheroid development, and stem-like cell proliferation in U87MG and U251 glioma cells [158]. A clinical trial has shown that combining BTZ with current standard chemo-radiotherapy is tolerable in newly diagnosed GBM patients, with a median progression-free survival of 18 months and a median overall survival of 19.1 months [159]. Concurrent treatment of BTZ with TMZ can promote Th1-driven immunologic responses in a subset of patients, resulting in improved clinical outcomes [160].

### 5.2. Autophagy Inducers

#### 5.2.1. Everolimus

Everolimus, a rapamycin analog, is an mTOR suppressor that can inhibit tumor cell compartment formation and angiogenesis [161]. Everolimus has been shown to have active anticancer efficacy with moderate toxicity in preclinical and clinical trials [162]. A combination of everolimus and Ad-Delta-24-RGD (oncolytic adenovirus) can trigger ATG5 expression and autophagy in U87MG cells and substantially increase median survival, resulting in 80% of long-term survival of a U87MG-derived mouse intracranial tumor model [162]. Similarly, everolimus can reduce glioma cell proliferation, thus promoting autophagy. TMZ and everolimus cotreatment can increase TMZ sensitivity in glioma cells. Cell proliferation is inhibited and autophagy is promoted more effectively when everolimus and TMZ are used together than when TMZ is used alone [163]. A clinical trial has revealed mild toxicity when everolimus is combined with conventional chemo-radiation [164].

#### 5.2.2. Temsirolimus (CCI-779)

Temsirolimus (CCI-779) is an ester of sirolimus (rapamycin) that can act as a particular inhibitor of mTOR. It can decrease the synthesis of cell cycle regulatory proteins and inhibit the expression of VEGF by inhibiting HIF-1α [165]. Preclinical results have shown additive or synergistic effects of Temsirolimus coupled with IR or TMZ in recurrent GBM and that treatment with Temsirolimus monotherapy is correlated with radiographic improvement in T2 signal abnormality [166]. Meanwhile, temsirolimus is well tolerated in patients with recurrent GBM based on clinical trials. It has been reported that 36% of temsirolimus-treated patients exhibit radiographic improvement, which is attributed to a slightly longer autoimmune thrombotic thrombocytopenic purpura (TTP). Elevated levels of phosphorylated ribosomal p70 S6 kinase (p70s6K) in baseline tumor samples tend to predict a patient population that will benefit from treatment with temsirolimus [167]. Additionally, in heavily pretreated adults with chronic malignant gliomas, a weekly dose of temsirolimus plus a daily dose of perifosine, an AKT inhibitor, is tolerable [168]. Meanwhile, clinical studies have shown that temsirolimus and HCQ cotreatment is safe and tolerable. Such cotreatment can modulate autophagy in patients, showing an anticancer potential [169].

#### 5.2.3. Metformin

Metformin (1,1-dimethyl-biguanide hydrochloride) is commonly used to reduce hepatic glucogenesis and improve insulin sensitivity and skeletal muscle glucose uptake in people with type 2 diabetes, [170]. Metformin has been linked to lower cancer incidence and mortality in several epidemiological studies. Its key anticancer molecular activity is linked to AMPK-mediated or AMPK-independent inhibition of mTORC1 which is involved in cancer cell metabolism, development, and differentiation [171]. According to preclinical studies, metformin can inhibit the mTOR pathway while simultaneously inhibiting cell proliferation and promoting cell cycle arrest, autophagy, and apoptosis in vitro. Cotreatment of metformin with IR or TMZ also can induce a synergistic antitumor response in glioma cell lines [172]. A combination of metformin and chemoradiation can modulate apoptosis, resulting in a higher Bax/Bcl-2 ratio and a lower ROS intake [173]. Meanwhile, combining metformin and TMZ can improve the reduction of phosphorylation of mTOR, 4EBP1, and S6K [174]. On the other hand, cotreatment of metformin with arsenic trioxide can help GSCs differentiate into nontumorigenic cells. Metformin works by activating the AMPK-FOXO3 axis, whereas arsenic trioxide inhibits the phosphorylation of STAT3 caused by IL-6 [175].

#### 5.2.4. Perifosine

Perifosine is an alkyl phospholipid that can inhibit AKT phosphorylation and activation by interfering with its recruitment to the plasma membrane. This drug can also inhibit ERK1/2 by activating c-Jun NH2-terminal kinase (JNK) and p21, causing G1/G2 cell cycle arrest [176]. Perifosine administration can destabilize many signaling pathways and promote the inhibition of cell proliferation in TMZ-treated GBM. In U87MG and U251MG GBM cells, perifosine can inhibit AKT/mTOR activation and cause mild apoptotic cell death. Perifosine’s cytotoxic activity in GBM cells is substantially improved when it is paired with short-chain cell-permeable ceramide (C6) [177]. Perifosine can also boost Bevacizumab’s proapoptotic effects and reverse Bevacizumab’s upregulation of phospho-AKT and MMP2 levels in C6 xenografts, resulting in better antiproliferative activity, reduced tumor development, and prolonged survival [178]. In heavily pretreated adults with recurrent malignant gliomas, a clinical trial has found that combining temsirolimus (115 mg/week) and perifosine (100 mg/day) (following a 600 mg load) is tolerable [168].

#### 5.2.5. Suberoylanilide Hydroxamic Acid (SAHA)

SAHA can inhibit the HDAC protein family by directly interfering with the enzyme’s catalytic site. Histone acetylation may improve the efficacy of anticancer drugs that target DNA [179]. According to previous research, SAHA can stop glioma cells from invading and reorganizing their TME, which is common after invasion [180]. Furthermore, SAHA can inhibit tumor growth and induce autophagy by increasing the formation of intracellular acidic vesicle organelles, attracting LC3-II to autophagosomes, potentiating levels of the BecN1 gene, and lowering levels of SQSTM1. Downregulation of AKT/mTOR signaling can cause autophagy to be activated. SAHA can speed up apoptosis and cause cell death [181]. SAHA also can cause apoptosis of GSCs and activation of caspase-8 and -9 mediated pathways. Interestingly, a lower dose of SAHA can inhibit GSCs by activating cell cycle arrest and causing premature senescence through p53 and p38 induction [182].

#### 5.2.6. Imipramine

Imipramine (3-(10,11-Dihydro-5H-dibenzo[b,f]azepin-5-yl)-N, N-dimethyl propane-1-amine) is a tricyclic antidepressant (TCA) that can inhibit 5-hydroxytryptamine (serotonin) and norepinephrine reuptake. It is used to treat serious chronic depression. In addition to its antidepressant properties, studies have shown that it can induce apoptosis in malignant tumor cells and elicit autophagic responses in primary astrocytes and neurons [183]. Imipramine can impede PI3K/AKT/mTOR signaling in U87MG cells, which decreases clonogenicity and promotes cell death. Furthermore, imipramine contributes to the development of acidic vesicular organelles, the conversion of LC3-I to LC3-II, and the redistribution of LC3 to autophagosomes, implying that it can promote autophagy progression. Imipramine-induced cell death can be prevented by the siRNA knockdown of Beclin-1 [183]. Meanwhile, imipramine can suppress the ability of U87MG and GBM8401 cells to invade and migrate by inducing apoptosis via extrinsic and intrinsic pathways. Furthermore, inhibiting the ERK/NF-κB pathway by imipramine can effectively block GBM progression in vivo [184]. Complementing imipramine with anticoagulant ticlopidine, which blocks the purinergic receptor P2Y12, can stimulate autophagy-associated cell death in glioma cells via the EPAC branch of the cAMP-signaling pathway, thus lowering cell viability in culture (LN229, LN71, LN443) and improving survival in glioma-bearing mice. ATG7 knockdown can eliminate the combinatorial effect of imipramine and ticlopidine, implying cell-lethal autophagy [185]. Imipramine and ticlopidine are two repurposed drugs approved by the US Food and Drug Administration (FDA) that can be used to amplify autophagy [186].

### 5.3. MiRNAs

Micro-RNAs (miRNAs) are endogenously expressed 18–25 nucleotide noncoding RNAs. By binding to target messenger RNA (mRNA), these molecules may inhibit gene expression by causing translational silencing or degradation. Cell proliferation, metastasis, cell death, angiogenesis, and drug resistance are only a few pathways that miRNAs can modulate [187]. In GBM cell lines U87MG and U251, as well as SHG44, upregulated miR cluster (MC-let-7a-1-let-7d) or STAT3 knockdown can suppress cell growth, promote cell death, and cause autophagy [188]. By specifically targeting Beclin-1 and preventing autophagy, miR-30a can increase the chemosensitivity of U251 GBM cells to TMZ [189]. Similarly, miR-93 can regulate autophagic activity in GSCs by simultaneously inhibiting multiple autophagy regulators, including ATG4B, ATG5, BECN1/Beclin-1, and SQSTM1/p62, thus enhancing IR and TMZ activity against GSCs [190]. MiR-128 overexpression can promote glioma cell apoptosis in GBM cells by activating caspase-3-9, degrading poly-(ADP ribose) polymerase, producing ROS, lowering mitochondrial membrane potential, and inducing nonprotective autophagy through mTOR inhibition [191]. In TMZ resistant U87MG cells, miR-519a can significantly increase TMZ-induced autophagy and apoptotic cell death, while miR-519a inhibition promotes TMZ resistance and reduces TMZ-induced autophagy. In addition, miR-519a can induce autophagy by modifying STAT3 expression. By promoting autophagy and targeting the STAT3/Bcl-2/Beclin-1 pathway, miR-519a can increase apoptosis and sensitize GBM to TMZ treatment in vivo [93].

### 5.4. Natural Products

Resveratrol (3,5,4′-trihydroxystilbene), a polyphenolic compound contained in nuts, grape skin, and red wine, can inhibit tumor initiation and progression; promote cell cycle arrest, apoptosis, and autophagy; and sensitize cancer cells to TRAIL by enhancing apoptosis and penetrating the BBB [192]. Resveratrol can inhibit the growth and infiltration of U87MG cells and GSC in vitro possibly through the deactivation of AKT and induction of p53. It can also suppress GBM development in vivo [193]. A combination of resveratrol and TMZ can trigger cell cycle arrest in the G2/M phase and increase reactive oxygen species (ROS), which acts as an upstream signal for AMPK activation. Activated AMPK can decrease mTOR signaling and downregulate Bcl-2, contributing to additive antiproliferative effects of the combination therapy in a GBM cell line SHG44 [194].

*Curcuma longa* (turmeric) rhizome contains polyphenol curcumin (1,7-bis (4-hydroxy-3-methoxyphenyl)-1,6-heptadiene-3,5-dione), also known as diferuloylmethane [195]. Curcumin can inhibit development; induce cell cycle arrest, apoptosis, and autophagy; inhibit angiogenesis and invasion; repress GSC proliferation; and induce differentiation and apoptosis of GSCs [196,197]. Curcumin can inhibit several tumor-specific pathways, such as NF-κB, STAT3, PI3K/AKT, and Sonic Hedgehog, allowing it to selectively destroy tumor cells while sparing healthy tissues. Resveratrol can increase TMZ/curcumin efficacy in brain-implanted tumors by blocking ERK1/2-mediated TMZ and curcumin-induced autophagy [198]. Curcumin can also induce GSC differentiation (SU-2 and SU-3) both in vivo and in vitro by inducing autophagy and suppressing tumor formation when GSCs are implanted intracranially into mice [199]. In GL261, U87MG, N2a, and F98 cells, treatment with curcumin and solid lipid curcumin particles can increase autophagy, decrease mitophagy markers, and inhibit the PI3K/AKT/mTOR pathway. Additionally, markers for cell survival are downregulated while markers for cell death are upregulated [200].

Dihydroartemisinin (DHA) is the main active metabolite of artemisinin. It can prevent the growth of several human cancers. It also can cross the BBB. In rat C6 glioma cells, DHA can increase ROS production and suppress the activity of glutathione-S-transferase, promoting radiosensitivity and potentiating the cytotoxic effect of TMZ. Combined treatment of DHA and TMZ can prevent the proliferation of U251, SKMG-4, T98G, and U373 glioma cells in vitro and improve the tumor inhibition efficacy of TMZ in vivo by inducing autophagy [201]. On the other hand, 9-tetrahydrocannabinol (THC) and other cannabinoids have anticancer effects in xenograft models. Oral administration of Sativex-like extracts with TMZ can provide a significant anticancer advantage in both intracranial and subcutaneous glioma cell-derived tumor xenografts [202]. A previous study has found that lethal mitophagy causes cannabidiol’s antitumor effect in glioma and identified that transient receptor potential cation channel subfamily V member 4 (TRPV4) is a target for mitophagy initiation [203]. Carnosic acid (CA), a natural benzenediol abietane diterpene found in *Rosmarinus officinalis* and *Salvia officinalis*, can also boost TMZ’s ability to inhibit colony formation and migration of GBM cells. By inhibiting cyclin B1 and activating PARP and caspase-3, CA and TMZ cotreatment can cause cell cycle arrest and apoptosis and promote cellular autophagy through p62 downregulation and LC3-I to LC3-II transfer [97]. 

Meanwhile, β-asarone can disrupt the cell cycle at the G0/G1 phase which inhibits the proliferation of U251 glioma cells. Beta-asarone can also stimulate autophagy through p53/AMPK/mTOR and p53/Bcl-2/Beclin-1 signal pathways [204]. Magnolin, a bioactive compound found in Magnolia plants, is commonly used in Eastern medicine to treat human diseases such as nasal congestion, sinusitis, emphysema, and inflammation. It has been shown that magnolin can inhibit TNF expression. Previous studies have also shown that magnolin can inhibit ERK and influence nitric oxide (NO) and prostaglandin E2 production [205]. Magnolol can inhibit LN229 and U87MG cell migration by decreasing the expression of focal adhesion-related proteins and N-cadherin [206].

Honokiol, a bioactive polyphenol derived from the Chinese herb Houpo, has anti-inflammatory, antimicrobial, anxiolytic, and antithrombotic properties. It can also slow the growth of glioma cells by activating the p53/cyclin D1/CDK6/CDK4/E2F1-dependent pathway which induces tumor cells to apoptosis and cell cycle arrest [207]. It can also enhance the expression of LC3-II and cleaved caspase-3 in mice with an intracranial glioma. Honokiol can also induce autophagy in U87MG cells in a dose- and time-dependent manner. The addition of 3-MA and rapamycin to the honokiol-induced autophagy pathway may lead to apoptosis by controlling the p53/PI3K/AKT/mTOR signaling pathway [208]. Thymoquinone (2-isopropyl-5- methylbenzo-1, 4-quinone) (TQ), a naturally occurring quinone, is the primary bioactive component of *Nigella sativa*’s volatile oil. It has anti-inflammatory, antineoplastic, and antioxidant properties. TQ can increase TMZ’s anticancer activity by inhibiting autophagy at the transcriptional stage, promoting apoptosis, and lowering the U87MG cell line’s colony-forming capacity and NO development [209]. Similarly, Euphol, the tetracyclic triterpene alcohol present in Euphorbia sap, has a concentration-dependent cytotoxic effect on GBM cells with a higher selective cytotoxic index than TMZ. Euphol and TMZ cotreatment can inhibit cell proliferation and motility. BafA1 treatment can increase LC3-II expression and the development of an acidic vesicular organelle, resulting in autophagy-associated cell death [210].

Rutin (vitamin P), a flavonoid found in a variety of edible plants, including green tea, onion, lemon, and apple, is known for its antioxidant, antidiabetic, anti-inflammatory, neuroprotective, and anticancer properties. Rutin and TMZ cotreatment can improve TMZ efficacy in a dose-dependent manner. TMZ can stimulate JNK development, leading to defensive autophagy, which is inhibited by rutin, resulting in increased apoptosis and decreased autophagy [211]. Similarly, vitamin D (VD) (1,25-dihydroxycholecalciferol), a hormonally active form, can induce autophagy by increasing cytoplasmic Ca^2+^ concentration. The Ca^2+^/calmodulin-dependent kinase is then activated, followed by AMPK activation. In a C6 GBM cell line, cytotoxic autophagy can increase antitumor activity when TMZ and VD are combined. In an orthotopic xenograft model, cotreatment with TMZ and VD can prevent tumor progression and extend animal survival [212].

## 6. Conclusions and Future Perspective

Glioblastoma multiforme (GBM) is the most common, heterogeneous, and highly invasive form of adult brain tumor with a poor median survival of approximately 15 months after diagnosis. Surgical resection accompanied by chemoradiation is the standard treatment for GBM patients. Despite decades of studies about its biology and therapeutic advancements, GBM recurrence remains a problem. Thus, more studies on GBM tumorigenesis and innovative treatments are required to improve the outcome of patients with GBM. Several studies have identified the role of autophagy in cancer tumorigenesis, including GBM pathogenesis, in recent years. Autophagy is a well-preserved evolutionary mechanism and an essential metabolic process that degrades and recycles cellular components to maintain homeostasis. Ironically, autophagy plays a dual function in GBM and other cancers as it can either suppress tumorigenesis by inhibiting cancer cell survival and causing cell death or promote tumorigenesis by encouraging metastasis and tumor development. Its upregulation can help cells survive treatment modalities, resulting in therapeutic resistance. Thus, autophagy modulation offers promising potential for developing novel and successful drugs for GBM treatment because of its complex and dual roles in cancer pathogenesis.

Glioma stem cells (GSCs) play a key role in glioblastoma treatment resistance and recurrence. The stemness of GSCs is enhanced by alterations in Wnt/β-catenin, mTOR, Notch, and Sonic Hedgehog molecular pathways, which can increase the proliferation, self-renewal, and pluripotency of GSCs [213]. Studies have shown that autophagy plays an important role in GSCs stemness features and GBM malignancy. Additionally, autophagy can attenuate normal neural stem cell (NSC) proliferation and differentiation as well as NSC niche maintenance as a major orchestrator of protein degradation and turnover, while its failure may lead to GSC maintenance and expansion [213]. Therefore, a better understanding of the molecular mechanism underlying autophagy’s effects on treatment resistance and GSCs could lead to better treatment outcomes for GBM. Moreover, given the complexity of the tumor microenvironment, more in vivo studies are needed to better understand how it interacts with autophagy in tumors.

On the other hand, selective autophagy is a specialized type of autophagy that protects cells from oxidative and genotoxic stresses by specifically targeting protein aggregates, defective or damaged organelles, and intracellular pathogens [31]. However, the exact functions of different types of selective autophagy (mitophagy, aggrephagy, ribophagy, reticulophagy, nucleophagy, xenophagy, lysophagy, lipophagy, ferritinophagy) have not been completely explored in GBM. Thus, unraveling the molecular mechanisms that regulate selective autophagy will be of great importance to better understand its suppressive and oncogenic role in gliomagenesis. Additionally, understanding the specific functions of these various types of selective autophagy in tumor growth provides an opportunity to more precisely target these pathways selectively. Lastly, modulating the activation of selective forms of autophagy that destroy metabolically important structures such as mitochondria, nuclei, and proteins in GBM cells may be exploited to impede tumor survival.

In recent years, autophagy inhibitors and inducers have been developed with promising results for autophagy modulation. Preclinical trials of autophagy inhibitors and inducers in conjunction with chemotherapeutic drugs may boost their efficacy and therapeutic outcomes. However, clinical trials on autophagy control are limited to only a few autophagy promoters and inhibitors. In addition, they have mixed results in terms of improving patient survival. As a result, further research is needed to determine the anticancer efficacy of treatment formulations that include autophagy modulators and chemotherapeutic drugs for GBM patients. Additionally, more research must be conducted to develop successful cancer-specific delivery systems and drugs to make tumors more responsive to treatments.

## Figures and Tables

**Figure 1 cells-10-01411-f001:**
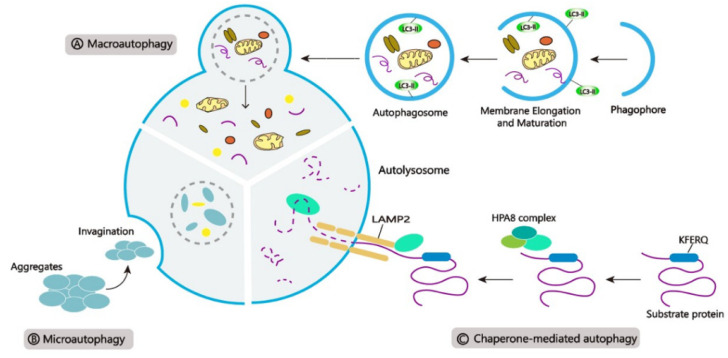
Three main types of autophagy. (**A**) Macroautophagy, (**B**) Microautophagy, and (**C**) Chaperone-mediated autophagy (CMA).

**Figure 2 cells-10-01411-f002:**
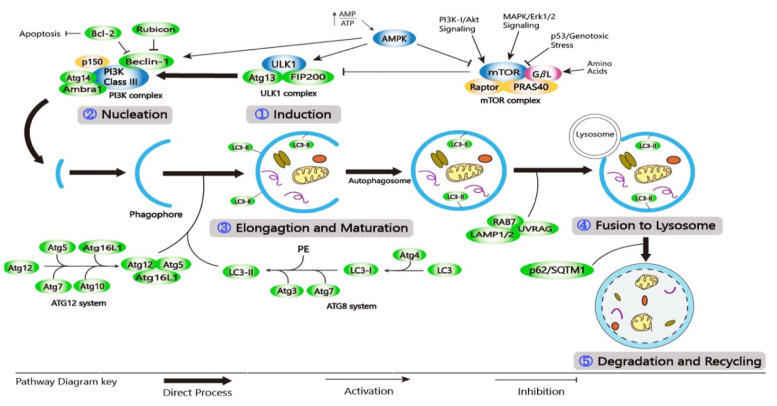
A schematic diagram of the molecular pathway of autophagy. (**1**) Induction; (**2**) Nucleation; (**3**) Elongation and Maturation; (**4**) Fusion to Lysosome; and (**5**) Degradation and Recycling.

**Figure 3 cells-10-01411-f003:**
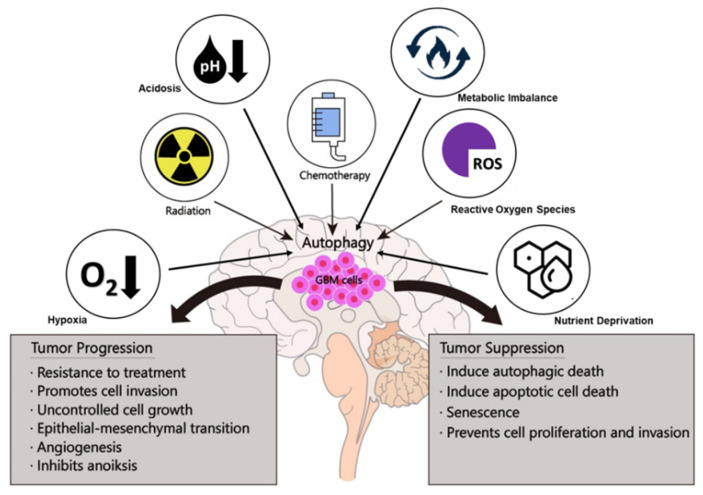
Dual functions of autophagy in GBM tumorigenesis.

**Table 1 cells-10-01411-t001:** Comparison of the different types of selective autophagy.

Pathway	Target	Physiological Function	Receptor	Ref
Aggrephagy	Misfolded protein aggregates	Removes misfolded protein aggregates	p62, NBR1, OPTN, TOLLIP	[30]
Mitophagy	Mitochondria	Removes weakened or nonfunctioning mitochondria	p62, NDP52, OPTN, NBR1, TAX1BP1	[31]
Reticulophagy	Endoplasmic reticulum	Controls ER morphology, turnover, ER luminal proteostasis, and recovery from stress	p62, OPTN, NBR1, BNIP3, RETREG1/FAM134B, FAM134B, RTN3, SEC62, ATL3, CCPG1, TEX264	[30,32,33]
Pexophagy	Peroxisome (Ub)	Degrades peroxisomes in response to the scarcity of nutrients which generates a high level of reactive oxygen species	SQSTM1/p62, NBR1	[28,31]
Lysophagy	Lysosome (Ub)	Degrades damaged or ruptured lysosomal membranes to prevent inflammation and cell death due to lysosomal contents leakage	SQSTM1/p62, NDP52 TAX1BP1, TRIM16	[34]
Xenophagy	Bacteria (Ub), Viruses	Degrades cytoplasmic bacteria and viruses	NDP52, OPTN, SQSTM1/p62, TAX1BP1	[27,31]
Nucleophagy	Nucleus	Degrades nuclear components necessary for terminal differentiation of keratinocytes	ATG39	[31]
Ribophagy	Ribosome	Degrades ribosomes during starvation to generate nucleotides and nucleosides	NUFIP1	[30]
Lipophagy	Lipid droplets	Regulates lipid production in response to various cellular stressors, especially in the liver	SQSTM1/p62	[30]
Ferritinophagy	Ferritin	Degrades ferritin in the lysosome to supply free Fe^3+^ to the cell for the synthesis of metalloproteins (hemoglobin and cytochromes)	NCOA4	[28]

## Data Availability

Not applicable.

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
