# Peer review of "Friend or Foe: Paradoxical Roles of Autophagy in Gliomagenesis"

_cells, 2021, doi:10.3390/cells10061411_

Round 1
Reviewer 1 Report
The present review titled Friend or Foe: Paradoxical Roles of Autophagy in Gliomagenesis, the author summarized two parts,
The autophagic molecular mechanisms and their function in the promotion or suppression of GBM tumorigenesis; and therapeutic methods and approaches for modulating autophagy activation and inhibition in GBM therapy.
In the section of Types of Autophagy,
The author never mentioned other types of autophagy, such as,
Mitophagy; aggrephagy; ribophagy; reticulophagy; nucleophagy; xenophagy; lysophagy;lipophagy; ferritinophagy.
Already have publications about those kinds of autophagy and glioblastoma cells.
ATF4 links ER stress with reticulophagy in glioblastoma cells
Glioma: Repurposed drugs combined to amplify autophagy
Those sections all were summarized quite good.
The Autophagic Pathway
Autophagy Regulation
Transcriptional Regulators of Autophagy
Autophagy inhibitors
Autophagy Inducers
The section of Conclusions and Future Perspective should rewrite,
A study published in Cancer Cell reports that two drugs approved by the US Food and Drug Administration (FDA)- the tricyclic antidepressant (TCA) imipramine and the anticoagulant ticlopidine- synergize to promote autophagy and slow glioma progression in mice. As the clinical use of TCAs has previously been associated with a reduced incidence of glioma, Hanahan and colleagues investigated the effects of imipramine on survival and tumour progression in genetically engineered mouse models of glioma. Imipramine treatment significantly prolonged survival in mice with low-grade glioma but had no such effect in mice with more progressive disease.
The author also should show us future perspective on other types of autophagy in Gliomagenesis. Such as, Mitophagy; aggrephagy; ribophagy; reticulophagy; nucleophagy; xenophagy; lysophagy;lipophagy; ferritinophagy.
Six things the author should keep in mind when writing a review article.
A review is not a list of results, Have a clear idea in mind about the structure you want for your article.
Reviewer 2 Report
The manuscript by Batara et al., entitled “Friend or Foe: Paradoxical Roles of Autophagy in Gliomagenesis”, represents an extensive review of the studies which investigated the role of autophagy in glioblastoma. The main mechanisms involved in the autophagy-dependent regulation of GBM cells are reported, along with potential therapeutic interventions targeting the autophagy pathway.
-The object of the manuscript is very ambitious, since it should include a huge studies, both in experimental models and in human patients. However, this aim was not fully accomplished, as it appears from a uncomplete citation of the previous literature. For instance, several important studies which demonstrate the involvement of autophagy in determining the stemness, invasive potential and other phenotypical and ultrastructural properties of GBM cells should be reported. Other studies carried out in xenograft are fundamental to comprehend the mechaisms involved in the brain invasion and relapse. All these studies, carried out in last decade, should be reported. Moreover, it is mandatory to cite the Guidelines for the use and interpretation of assays for monitoring autophagy (2021), when the autophagy pathway along with autophagy-related organelles are described.
-Concerning the role of autophagy in promoting or counteracting GBM, it should be useful to report in a separate table the main studies which demonstrated such a dual role of the autophagy pathway.
- Conclusions are not effective in providing an unifying view, showing the actual perspective of how the autophagy pathway may be useful to comprehend the neurobiology of the GBM.
- Paragraph 3.1 AMPK/mTORC1/ULK1 pathway. What is the “autolysosome”? Maybe the Authors refer to the autophagolysosome
Many thanks.
Reviewer 3 Report
The Review titled “Friend or Foe: Paradoxical Roles of Autophagy in Gliomagenesis” gives a comprehensive understanding of the pleiotropic functions of autophagy in the GBM.
In consideration of the promising role of autophagy in GBM treatment, the authors need to add the following references:
-Ryskalin L, Gaglione A, Limanaqi F, Biagioni F, Familiari P, Frati A, Esposito V, Fornai F. The Autophagy Status of Cancer Stem Cells in Gliobastoma Multiforme: From Cancer Promotion to Therapeutic Strategies. Int J Mol Sci. 2019 Aug 5;20(15):3824. doi: 10.3390/ijms20153824. PMID: 31387280; PMCID: PMC6695733.
- Arcella A, Biagioni F, Antonietta Oliva M, Bucci D, Frati A, Esposito V, Cantore G, Giangaspero F, Fornai F. Rapamycin inhibits the growth of glioblastoma. Brain Res. 2013 Feb 7;1495:37-51. doi: 10.1016/j.brainres.2012.11.044. Epub 2012 Dec 19. PMID: 23261661.
Ryskalin L, Lazzeri G, Flaibani M, Biagioni F, Gambardella S, Frati A, Fornai F. mTOR-Dependent Cell Proliferation in the Brain. Biomed Res Int. 2017;2017:7082696. doi: 10.1155/2017/7082696. Epub 2017 Nov 13. PMID: 29259984; PMCID: PMC5702949.
Ferrucci M, Biagioni F, Lenzi P, Gambardella S, Ferese R, Calierno MT, Falleni A, Grimaldi A, Frati A, Esposito V, Limatola C, Fornai F. Rapamycin promotes differentiation increasing βIII-tubulin, NeuN, and NeuroD while suppressing nestin expression in glioblastoma cells. Oncotarget. 2017 May 2;8(18):29574-29599. doi: 10.18632/oncotarget.15906. PMID: 28418837; PMCID: PMC5444688.
Ryskalin L, Biagioni F, Busceti CL, Lazzeri G, Frati A, Fornai F. The Multi-Faceted Effect of Curcumin in Glioblastoma from Rescuing Cell Clearance to Autophagy-Independent Effects. Molecules. 2020 Oct 20;25(20):4839. doi: 10.3390/molecules25204839. PMID: 33092261; PMCID: PMC7587955.
Ferese R, Lenzi P, Fulceri F, Biagioni F, Fabrizi C, Gambardella S, Familiari P, Frati A, Limanaqi F, Fornai F. Quantitative Ultrastructural Morphometry and Gene Expression of mTOR-Related Mitochondriogenesis within Glioblastoma Cells. Int J Mol Sci. 2020 Jun 27;21(13):4570. doi: 10.3390/ijms21134570. PMID: 32604996; PMCID: PMC7370179.
Chen Y, Li N, Wang H, Wang N, Peng H, Wang J, Li Y, Liu M, Li H, Zhang Y, Wang Z. Amentoflavone suppresses cell proliferation and induces cell death through triggering autophagy-dependent ferroptosis in human glioma. Life Sci. 2020 Apr 15;247:117425. doi: 10.1016/j.lfs.2020.117425. Epub 2020 Feb 11. PMID: 32057904.
Shahcheraghi SH, Zangui M, Lotfi M, Ghayour-Mobarhan M, Ghorbani A, Jaliani HZ, Sadeghnia HR, Sahebkar A. Therapeutic Potential of Curcumin in the Treatment of Glioblastoma Multiforme. Curr Pharm Des. 2019;25(3):333-342. doi: 10.2174/1381612825666190313123704. PMID: 30864499.
Round 2
Reviewer 1 Report
The response letter wrote quite well , however ,when we come back in the manuscript ,it is a quite different situation, for example,
- In the section of Types of Autophagy, the author never mentioned other types of autophagy, such as,
Mitophagy; aggrephagy; ribophagy; reticulophagy; nucleophagy; xenophagy; lysophagy; lipophagy;
ferritinophagy. There are already publications about those kinds of autophagy and glioblastoma cells. Eg.
ATF4 links ER stress with reticulophagy in glioblastoma cells, Glioma: Repurposed drugs combined to
amplify autophagy. Those concerns still not well mentioned. I request the author have a table for all kinds of autophagy in the section of (Types of Autophagy). - 5. The author also should show us future perspective on other types of autophagy in Gliomagenesis. Such as,
Mitophagy; aggrephagy; ribophagy; reticulophagy; nucleophagy; xenophagy; lysophagy; lipophagy;
ferritinophagy.Those concerns still not well mentioned. - Anyway , for the manuscript , the editor and reviewer need very clear version ,you just need label in color which you revised, don't need to submit your messy version.
Reviewer 2 Report
The Authors carried out substantial changes according with the reviewer's suggestions which improved the manuscript.
Author Response
Thank you very much. We appreciate the time and effort you have dedicated to providing us with insightful feedback, and inputs for the improvement of our manuscript.
This manuscript is a resubmission of an earlier submission. The following is a list of the peer review reports and author responses from that submission.